# Brucellosis in Humans and Animals in Kyrgyzstan

**DOI:** 10.3390/microorganisms10071293

**Published:** 2022-06-25

**Authors:** Kalysbek Kydyshov, Nurbolot Usenbaev, Almaz Sharshenbekov, Narynbek Aitkuluev, Murat Abdyraev, Salamat Chegirov, Jarkynay Kazybaeva, Hanka Brangsch, Falk Melzer, Heinrich Neubauer, Mathias W. Pletz

**Affiliations:** 1Institute of Bacterial Infections and Zoonoses, Friedrich-Loeffler Institut, Naumburger Str. 96a, 07743 Jena, Germany; hanka.brangsch@fli.de (H.B.); falk.melzer@fli.de (F.M.); heinrich.neubauer@fli.de (H.N.); 2Republican Center for Quarantine and Highly Dangerous Infections of Ministry of Health and Social Development, Skrybina Str. 92, Bishkek 720005, Kyrgyzstan; usenbaev_nurbolot@mail.ru (N.U.); kazybaeva94@inbox.ru (J.K.); 3The State Veterinary Service under Ministry of Agriculture of the Kyrgyz Republic, Kievskaya Str. 96b, Bishkek 720033, Kyrgyzstan; zamlazi@mail.ru; 4The Republican Clinical Infectious Diseases Hospital, L. Tolstoy Str. 70, Bishkek 720017, Kyrgyzstan; n-aitkuluev@inbox.ru; 5Kyrgyz Scientific Research Institute of Veterinary Medicine “A. Duisheeva”, Togolok Moldo Str. 60, Bishkek 720040, Kyrgyzstan; abdyraev71@mail.ru (M.A.); ch_salamat15@mail.ru (S.C.); 6Institute of Infectious Diseases and Infection Control, Jena University Hospital, Am Klinikum 1, 07747 Jena, Germany; mathias.pletz@med.uni-jena.de

**Keywords:** brucellosis, epidemiology, vaccine, livestock production, animal, human, Kyrgyzstan

## Abstract

Brucellosis is a globally reemerging and neglected zoonosis causing serious public health problems as well as considerable economic losses due to infection of livestock. Although the epidemiology of brucellosis has been well studied and its various aspects in humans and animals are well understood, it is still one of the most challenging health problems in many developing countries such as Kyrgyzstan. This review describes epidemiological characteristics of brucellosis in humans and animals, its impact on animal production and the role of implemented infection control measures in Kyrgyzstan. Particularly, introduction of mass vaccination in small ruminants evidently contributed to control of brucellosis in Kyrgyzstan, reducing the number of infections in animals as well as humans.

## 1. Introduction

Brucellosis is a neglected zoonotic disease that causes substantial economic losses in animal production and severe acute and chronic infections in humans [1]. According to the World Health Organization (WHO) more than 500,000 new human cases are reported annually worldwide. Currently, the highest incidence of human brucellosis is recorded in the Middle East and Central Asia [2].

Brucellosis is caused by bacteria of the genus *Brucella*, which are facultative intracellular Gram-negative coccobacilli, non-spore-forming and non-capsulated, with reservoirs in farm animals and wildlife. Due to their high virulence and contagiousness, the main threat for public health emanates from three species: *Brucella abortus* (reservoir in cattle), *Brucella melitensis* (reservoir in sheep and goats) and *Brucella suis* biovars 1 and 3 (reservoir in pigs). These *Brucella* spp. cause severe clinical illness in humans [3,4].

In animals, brucellosis is highly contagious and cross-species transmission of *Brucella* spp. occurs frequently when host species are kept together in enclosed areas. The bacteria are mainly localized in the reproductive organs and the lymph nodes of their hosts. They are excreted in large quantities through urine, milk, placental, and other fluids. The clinical picture in animals varies depending on the host species. The incubation period lasts two to four weeks. In general, bovine brucellosis (*B. abortus*), caprine brucellosis (*B. melitensis*), and porcine brucellosis (*B. suis*) manifest as fever, mastitis, weak offspring, and infertility in both sexes. Spontaneous abortion is recognized as one of the most prominent symptoms of brucellosis [5].

Brucellosis is transmitted to humans mainly by consumption of unpasteurized milk or through direct contact with infected animals, particularly with secretions, placenta, and aborted fetuses during delivery. In humans, the incubation period of the disease is usually one to four weeks, but can last up to two months. At the acute stage, undulant fever is a common symptom accompanied by night sweats, chills, fatigue, headache, and arthralgia. After replication in lymphatic tissue, the bacteria can spread via the blood stream and may also replicate in the kidney, liver, spleen, breast tissue, or joints, causing both localized and systemic infection. Any organ system can be involved (e.g., central nervous system, cardiovascular system, skeletal system, genitourinary system, pulmonary system, and skin). The localization of the infection may lead to focal symptoms (e.g., epididymo-orchitis, infertility, osteomyelitis, and endocarditis) [6]. Human brucellosis is treated with antibiotics, e.g., tetracycline or doxycycline in combination with other drugs such as rifampin or streptomycin.

Vaccination of cattle and small ruminants is an effective way to control outbreaks and minimize economic losses [5]. Clinical brucellosis is diagnosed indirectly by serological or other immunological tests and directly by isolation of brucellae, e.g., from blood, tissue samples, and stomach content of aborted fetuses [7].

In Kyrgyzstan, the prevalence of brucellosis in humans and animals is higher than in other Central Asian countries, which poses a serious public health and veterinary problem. Persistent traditional agricultural practice and lifestyles, as well as consumption of fresh dairy products, contribute to this high prevalence. Currently, despite its endemicity, brucellosis remains underdiagnosed and underreported. Also, the transmission between livestock and humans has not yet been studied in detail in Kyrgyzstan.

The objective of this review is to highlight the existing knowledge on the epidemiology of brucellosis in humans and animals and to review the factors associated with its persistence in Kyrgyzstan. This analysis will provide a scientific basis for policymakers for informed, evidence-based decisions to reduce the disease burden in both animals and humans. Involving all official institutions of the country engaged in the control of brucellosis, a comprehensive overview is provided which can serve as a blueprint for other zoonoses as well.

## 2. Materials and Methods

To analyze recent trends, data on brucellosis were collected from 2010 to 2020. We performed a literature search to identify articles on animal and human brucellosis in Kyrgyzstan. Data on animal production and corresponding regulations were obtained from the State Inspectorate for Veterinary and Phytosanitary Safety of the Kyrgyz Republic (SVPS) and data on infected people was gathered from the Republican Center for Quarantine and Highly Dangerous Infections Ministry of Health and Social Development (RCQHDI). In addition, data from the World Health Organization (WHO), the Food and Agriculture Organization of the United Nations (FAO), and the World Organization for Animal Health (OIE) were used. Data from governmental sources were provided by the National Statistical Committee of the Kyrgyz Republic and the Ministry of Justice of the Kyrgyz Republic and Ministry of Agriculture of the Kyrgyz Republic.

The literature search was performed in international repositories (Google Scholar, PubMed, Web of Science and Elibrary.ru) using following search terms “Brucellosis”, “Brucella”, “human and animal brucellosis”, “Brucellosis AND Kyrgyzstan” and “Prevalence OR Incidence OR Risk AND Factors”. In addition, the Russian-language version of the Elibrary.ru database was used to identify articles of interest only available in Russian.

In total, 12 relevant research articles were found: seroprevalence surveys (livestock and humans) (*n* = 2), case control study (*n* = 3), economic impact study (*n* = 1), antibiotic susceptibility study (*n* = 1), molecular epidemiology (*n* = 5).

## 3. Country Profile

The Kyrgyz Republic (Kyrgyzstan) is located in the North-East of Central Asia. The territory of Kyrgyzstan is 900 km from West to East, 410 km from North to South and has borders in the North with Kazakhstan (1113 km), in the West and North-West with Uzbekistan (1374 km), in the South-West with Tajikistan (972 km) and in the South and Southeast with China (1049 km). The total area of pastures and hayfields is 9,147,000 hectares, more than 85% of the total area of agricultural land (personal communication Abdyraev M. (principal adviser, scientist, the Kyrgyz Research Institute of Veterinary Medicine A. Duisheev (KRIV); on 3 January 2022).

According to the State Inspectorate for Veterinary and Phytosanitary Safety of the Kyrgyz Republic, Kyrgyzstan has a population of 6.64 million people, 4.36 million of them live in rural areas. Approximately 40% of the population is traditionally employed in agriculture and animal husbandry. The country is divided into seven administrative oblasts (regions Batken, Chuy, Jalal-Abad, Naryn, Osh, Talas, and Issyk-Kul. Livestock raising, cultivation of cotton, fruits, vegetables, grain crops, tobacco, and wool production are the leading branches of agriculture. Livestock comprises mainly sheep and goats, as well as dairy and beef cattle including yaks. Horses are used as means of transport, source of meat and for koumiss, as well as fermented mare’s milk, which is regularly consumed and also used for medicinal purposes [8].

Wildlife is insufficiently registered in Kyrgyzstan. There are some rare species living in the mountains at 3000–5000 m above mean sea level (MSL), such as the argali sheep (Arkhar (*Ovis vignei*), Marco Polo or Pamir argali (*Ovis ammon polii*)*;* the Tien Shan argali (*Ovis ammon karelini* and the Nuratau argali (*Ovis ammon severtzovi*)), the Sibirian ibex (*Capra sibirica*) or the snow leopard (*Panthera uncia*). The Tien Shan maral (*Cervus canadensis songaricus*) and *Capreolus pygargus*, red and roe deer, represent *Cervidae*. Frequent predators are foxes, wolves, and bears. Many of these species are listed in the Kyrgyz red lists for endangered animals but may also play a role as wildlife reservoirs for brucellosis [9,10].

## 4. Livestock and Animal Production

In Kyrgyzstan, livestock and their products are major components of the agriculture sector playing an important role in the national economy. Sheep breeding has a long tradition and adds large parts to the country’s economic returns. Most livestock are owned by private households and small farmers in rural areas who earn an income by trading wool, animals, and milk (personal communication Sotovaldiev A., principle adviser, Department of Epizootic Surveillance, the State Inspectorate for Veterinary and Phytosanitary Safety of the Kyrgyz Republic (SIVPS); on 3 January 2022).

An average altitude of 3000 m MSL and about 9.1 million ha of natural pastures make transhumance the most important livestock production system in the country. Traditionally pastures of the highlands (2000–3000 m MSL) are used in summer and those of the lowlands (1000–1500 m MSL) in winter (Figure 1). During the autumn and spring seasons (November until mid-April), animals are generally kept nearby the stables, graze in the river valleys below the forest zone, and have access to supplementary feed at the stables where they stay during the night. The feed for the animals in the stables consists of crop residues, hay or silage from alfalfa, grass or legumes as well as some silage for cattle from cereal crops. The herds belong to various owners and the different farm animal species mingle uncontrolled. In mid-spring, “community herd shepherds” gather animals from private household-livestock owners of the villages for migration to high mountain pastures later on in early summer. Slow migration begins in early summer to highlands where the mixed herds stay on the pastures during June and August and return to lowland valleys until mid of October [11].

In recent years, the number of farm animals has gradually increased, i.e., from 2016 to 2020 the number of cattle increased by 12.3%, sheep and goats by 4.3%, horses by 15.5% and poultry by 7.0%. Livestock inventory data from official reports show that the numbers of small ruminants and poultry are approximately six million (Table 1). In contrast, the number of domestic pigs is low, probably due to the high proportion of Muslim population.

Milk, beef, and mutton are traditionally the most important elements of the diet of Kyrgyz people. In 2020, 1,668,000 tons of milk and 230,400 tons of meat were produced [14]. Most animals are sold on animal markets, and most of the meat is sold through the bazaar system. After Kyrgyzstan joined the Eurasian Economic Union (EAEU) in 2015, Kazakhstan and Russia became the most accessible markets. In recent years, the export of animals and meat has been growing, with Kazakhstan, Tajikistan, Uzbekistan, and Russia, as well as the Persian Gulf countries (Iran, Kuwait, Qatar, and the UAE) being the major export destinations. According to the UN Comtrade Database, in 2020, Kyrgyzstan exported sheep, goats, and bovines to Uzbekistan, Tajikistan, and Kazakhstan worth $5 million and $14 million, respectively. In the same year, 91,159 tons of mutton and goat meat were sold, mainly to the United Arab Emirates, Kuwait, and Uzbekistan (Table 2). Exports of mutton and goat meat to Iran are estimated to be more than one million tons of meat worth $5 million from 2016 to 2019. Overall, the value of animal exports continues to exceed meat exports. Kyrgyzstan has decreased the import of meat products in the last four years. Belarus and the Russian Federation are the main countries from which meat is imported [15]. China is Kyrgyzstan’s largest import trading partner and the fastest growing import market over the past 10 years [16]. However, export and import of animals and meat between the two countries is under-developed.

Milk and dairy products are considered to be the main prospective products that can increase the volume of exports [17,18,19]. According to the National Statistics Committee, the volume of exports of these products has doubled in 2020 compared to 2016 with an estimated amount of nearly 35,000 tons of milk worth $46 million (Figure 2). In 2020, dairy products were exported mainly to Kazakhstan, the Russian Federation, Uzbekistan, and Tajikistan worth $24 million, $21 million, $705,000, and $217,000, respectively [20]. In accordance with the technical regulations of the EAEU, meat and milk as well as their products are supplied to the domestic market and exported to trading partners (personal communication Sotovaldiev A., principal adviser SIVPS on 6 January 2022).

Over the past five years, Kyrgyzstan has imported more than 7.8 million tons of milk worth $6.5 million (Table 3), with Kazakhstan being the main supplier with 6 million tons of milk worth $4.6 million [15].

## 5. Veterinary Laws and Regulations for Livestock Production and Products in Kyrgyzstan

In Kyrgyzstan, livestock production as well as export and import of livestock and its products is one of the most important sectors of the economy. Table 4 summarizes the main relevant laws and regulations of the Kyrgyz Republic. These establish requirements for the safety of meat and its products as well as milk and dairy products regarding production processes, storage, transport, sale, disposal, but also labeling and packaging to protect the population from diseases common to humans, animals, and the environment. In general, animals and animal products are subject to compulsory veterinary and sanitary inspection in accordance with laws and regulations and must comply with veterinary and sanitary requirements for the production, sale, and slaughter of animals.

Export and import control of animals and livestock products is monitored by the State Inspectorate for Veterinary and Phytosanitary Safety of the Kyrgyz Republic (State Inspectorate). In order to maintain a Unified Register of Identified Animals, Adventel, a company based in France, developed the “System for Identification and Traceability of Animals” (SITA) in 2015, that is used to control the movement of animals (personal communication Sotovaldiev A., principal adviser, SIVPS).

## 6. Brucellosis in Animals

The identification of brucellosis-positive small ruminant flocks and cattle farms began in 1932 in Kyrgyzstan when veterinary laboratories and diagnostic methods were implemented. Cases of brucellosis in cattle and small ruminants are registered since 1940 [22]. Currently, brucellosis in livestock is registered mainly in cattle and small ruminants. Camel raising is not very popular in Kyrgyzstan and is not controlled/registered. However, several studies have proven that camels are very susceptible to *Brucella* and other bacterial pathogens [23]. Similarly, the presence of susceptible wildlife within the territory of the country is of particular importance in relation to brucellosis since epidemiological control of brucellosis in wildlife is not officially regulated.

According to the State Inspectorate for Veterinary and Phytosanitary Safety of the Kyrgyz Republic, serology remains the main diagnostic tool for brucellosis (e.g., Rose–Bengal plate test, serum agglutination test, Complement Fixation Test, enzyme-linked immunosorbent assay (ELISA) and milk ring test). Samples from abortions are sent to state laboratories and cultured. Use of PCR or phenotype identification of *Brucella* isolates has not yet been well established.

At rayon level, the Regional Veterinary Administration (RVA) works closely together with pasture management committees, village health committees, private veterinary units, public health sector and livestock owners. The RVA is also collecting animal census information at village level in collaboration with private veterinarians. This information is important for planning all animal disease control programs. Private veterinarians at the Ayil-Okmot level (the rural government administration superior to the village head) develop and provide the RVA with an annual plan and schedule for screening of livestock, stating the number of serum samples to be tested and the number of doses of vaccine needed.

From 2010 to 2020, official screening tests for the whole country showed that 0.35% of the cattle sera (10,874,642 sera tested) and 0.71% (864,057 sera tested) of small ruminants were positive. There were 179 horses and 12 dogs found to be seropositive in private households and farms. At the same time, the estimated prevalence was 0.23% in cattle and 0.01% in small ruminants. The number of tested animals was low compared to the total number of animals (Table 5). The annual average of tested animals was estimated at 714,424 cattle and 78,551 small ruminants (personal communication Sotovaldiev A. (principal adviser, SIVPS), Abdyraev M. (principal adviser, scientist, the Kyrgyz Research Institute of Veterinary Medicine A. Duisheev KRIV) on 11 Febuary 2022).

The percentage of seropositive livestock varies between different regions of Kyrgyzstan. For instance, the highest percentages in cattle were found in Naryn (0.64%), Issyk-Kol (0,51%), Chui (0.48%), and Bishkek (0.30%), followed by Talas (0.27%), Jalal-Abad (0.15%), Osh city (0.15%) and Batken (0.09%) over the past eleven years (Figure 3). During the same period, the percentages of seropositive small ruminants were 1.87%, 1.08%, 0.91%, 0.87%, and 0.86% in Talas, Batken, Jalal-Abad, Chui, and Issyk-Kul, respectively (Figure 4). These data demonstrate that brucellosis is endemic in cattle and small ruminants in all oblasts. However, there is no apparent correlation in the level of percentage seropositivity between these animal groups. For example, higher seropositivity in cattle was reported in Naryn, Issyk-Kol, Chui, and Bishkek, while in small ruminants the percentage was higher in Talas, Batken, Jalal-Abad, Chui, and Issyk-Kul.

A Kyrgyz-Swiss study conducted in 2006–2007 showed that seroprevalences were 3.3% in sheep, 2.5% in goats, 2.7% in cattle, and 8.8% in humans in Naryn, Chui, and Osh according to serological tests (Rose–Bengal test, Huddleson test and ELISA). Most of these oblasts are livestock breeding areas. In that study the oblast Naryn showed the highest seroprevalence in sheep which was often associated with human brucellosis. This investigation was followed by a study to compare the efficacy of six serological tests in cattle, small ruminants, and humans using a Bayesian model. The true seroprevalence of brucellosis in Kyrgyzstan was shown to be actually higher with 7%, 3%, 12%, and 15% in humans, cattle, sheep, and goats, respectively. In addition, the Rose–Bengal test has been confirmed as a useful screening test in cattle and humans [24,25].

Table 6 summarizes available studies on *Brucella* isolates from Kyrgyzstan. In this literature review, five studies were found in local scientific journals published in Russian and only one paper was found in an international journal. According to these studies, 427 *B. melitensis* strains were found in small ruminants and 16 in cattle, where 19 *B. abortus* strains were isolated from small ruminants and 11 from cattle. In 2013, Kasymbekov et al. reported the isolation of 17 strains of *B**. melitensis* from aborted fetuses of sheep (*n* = 15) and cattle (*n* = 2). Those were also the first isolates from Naryn. The study confirmed that *B. melitensis* is endemic in Naryn and sheep are apparently the main source of infection for cattle [27]. However, *Brucella* isolates from animals of other Kyrgyzstan regions are rare. Identification and genotyping of *Brucella* isolates are not routinely established. Therefore, the genetic epidemiology of circulating strains cannot be assessed in detail throughout the country.

## 7. Vaccination

Vaccination of livestock is one of the best strategies for controlling brucellosis. Figure 5 shows the timeline of vaccination programs in cattle and small ruminants in Kyrgyzstan. Specific prophylaxis was not carried out until 1952 when the Soviet Union began to prevent and control brucellosis with various vaccines based on *Brucella* sp. strains such as *B. abortus* S19, S82, 104M for cattle and *B. melitensis* Rev1, 38/59, Nevsky-12 for small ruminants [30]. Despite the *B. melitensis* Rev1 vaccination, there was a steady increase in brucellosis cases between 1977 and 1989 (Table 7), which could be attributed to ineffective administration of the vaccine and the lack of a large-scale brucellosis vaccination campaign. There was no detailed and consistent immunization schedule, and the *B. melitensis* Rev1 vaccine used did not meet international standards. In addition, the cold chain could not be guaranteed and not enough veterinarians could be recruited.

In 1992, vaccination of cattle and small ruminants with a locally produced Kyrgyz *B. melitensis* vaccine was initiated. However, this project was abandoned in 1994 due to lack of vaccination efficacy and for economic reasons. Mass vaccination of small ruminants with a *B. melitensis* Rev1-based vaccine began in 2008, funded by the World Bank and by several external projects. The vaccination program was performed annually in spring and autumn with live attenuated *B. melitensis* Rev1 (BRUCEVAC, Jordan Bioindustrial Center, JOVAC) applied via the conjunctival route. All adult male and all non-pregnant adult female sheep and goats, as well as lambs and kids between 4 and 8 months were vaccinated with a full dose (1 × 10^8^ bacteria) in the first year and replacement ewes in the spring and fall in subsequent years. In the framework of this new strategic plan, the vaccination of cattle against brucellosis with *B. abortus* S19 (JOVAC), also a live, attenuated vaccine [33], started in 2019. This *B. abortus* S19 vaccine is administered to all bovines (except bulls), heifers and calves between 3 and 18 months at a dose of 2 mL (50−100 × 10^9^ bacteria) subcutaneously in the first year and annually to all heifers between 3 and 18 months.

Between 2010 and 2020, 28,322,448 small ruminants were vaccinated. Since 2011, as a result of the implementation of the vaccination plan, the cases in cattle and small ruminants dropped to one third, coinciding with a decrease of 50% in human cases (Figure 6). The Kyrgyz Research Institute of Veterinary Medicine A. Duisheev (KRIV) is responsible for monitoring the vaccination quality. Samples are taken from randomly selected flocks and herds of vaccinated animals approximately 21 days after vaccination to control the performance and quality of each vaccination team. The data must be submitted to the RADIS (Rayon Animal Disease Information System) of the RVA and subsequently entered into the information system NADIS (National Animal Disease Information System).

## 8. Brucellosis in Humans

The Republican Center for Quarantine and Highly Dangerous Infections Ministry of Health and Social Development (RCQHDI) is the authority responsible for human brucellosis control in Kyrgyzstan. According to the RCQHDI, the epidemiological timeline of prevalence is divided into a first stage with low incidence (1950–1990) during the Soviet Union period, a second with high incidence after independence in 1991 (1991–2011), and a third stage with a decline in numbers of cases (2012 until today). RCQHDI estimates that 500–900 new cases of brucellosis have been reported annually in recent decades and the highest historical incidence of human brucellosis was 80 cases per 100,000 inhabitants in 2011 (Figure 7).

Human brucellosis control, diagnosis, data collection and patient follow-up are carried out in accordance with Order No. 586 of 10 August 2018 “on improving measures to prevent human brucellosis in the Kyrgyz Republic” [36]. In recent years, human brucellosis has been notified from all regions of the country. From 2010 to 2020, 18,279 new brucellosis cases were registered. The highest incidences were recorded from Jalal-Abad (55/100,000), Talas (51.4/100,000), Naryn (45.6/100,000), and Issyk-Kul (41.6/100,000) (Figure 8), which coincides with a high number of livestock. A 10-years analysis showed that all age groups were affected by brucellosis, with the majority (80%) of the cases occurring among active workers. In addition, 409 family outbreaks and 1990 cases of brucellosis in the age group <14 years were registered. However, cases in livestock and humans, especially rural residents, are not expected to be consistently diagnosed.

Clinical diagnosis includes the evaluation of clinical manifestations, serological tests, and epidemiological history of patients. After clinical and laboratory confirmation, each human case is investigated by an epidemiologist for surveillance purposes, demographic information is assessed as well as data on food consumption, contact with animals, and type of work or activity at the time of onset. In addition, according to Order No. 586, a joint investigation is conducted with veterinarians to identify the infection source, e.g., the patient’s livestock. From 2010 to 2020, a total of 6912 cases in small ruminants and 4396 cases in cattle were identified in this way as sources of brucellosis in humans. However, during the same period, 6671 cases were classified as “unknown source of infection” (Figure 9).

To identify risk factors for brucellosis, case-control studies in humans and herds or flocks used to be conducted [37]. During the past decade, only three case-control studies [38] were conducted in Osh, Batken, Haryn, and Jalal-Abad which showed consistently that contact with aborted animals, consumption of unpasteurized homemade cream, stable cleaning work, sheep shearing, and livestock at home were major risk factors [39,40,41].

From 2015 to 2020, 5087 acute brucellosis cases were reported with the highest number of cases (*n* = 1014) in 2017. Human brucellosis cases can occur throughout the year. On average, 70 cases per month were registered and the highest case numbers were reported for May, July, and June in 2017 with 153, 144, and 141 cases, respectively (Figure 10). The lowest number of reported cases was 19 in November 2020. During the same period, the peak of the epidemic season was from March to September, with the highest numbers of cases reported in May, June, and July. Overall, June was the month with the highest numbers of reported cases i.e., 654, followed by 650 cases in May and 594 cases in July. In contrast, the lowest number of cases was reported in January (209 cases). This pattern correlates well with the lambing season from February to March.

In 2021, the RCQHDI bacteriological laboratory was accredited by the International Organization for Standardization (ISO15189-2015), attesting its technical competence for brucellosis diagnosis. Currently, serologic testing is the main method used for humans, i.e., the Huddleson plate agglutination test [42]. The same antigenic reagent is used for the Wright test (“quantitative” test), which is subsequently applied in case of a positive Huddleson test. Although the RCQHDI is equipped with ELISA and conventional and RT-PCR units, no PCR-based detection of *Brucella* isolates or typing is conducted. Genus identification of *Brucella* strains is performed exclusively by colony morphology, aerobic or anaerobic growth, and Gram and Stamp staining methods.

## 9. Clinical Feature

Human brucellosis cases are classified into three groups according to clinical history, symptoms, and time span of clinical manifestation: acute brucellosis (0–2 months), subacute brucellosis (2–12 months), and chronic brucellosis (>12 months) [43]. Between 2010 and 2020, the RCQHDI has registered 15,806 acute, 750 subacute, and 1,723 chronic cases (*n* = 18,279 human cases). In acute brucellosis, blood samples show a strong positive Huddleson plate agglutination test and a high Wright test titer (1:200 or higher). According to the recommendations of the World Health Organization (WHO) and the Kyrgyz clinical protocol from 2005, patients with acute brucellosis are hospitalized and receive gentamicin and doxycycline for the first 14–21 days depending on the severity of disease. Administration usually results in a rapid clinical improvement. The remaining 31 treatment days with doxycycline are supervised by the family medicine service. Patients are discharged from the hospital when they make clinical progress, or when the Wright test titer has dropped repeatedly. Patients must also be seen by an infectious disease physician twice a year for the next two years. They are advised to avoid unpasteurized milk, homemade cream, and contact with animals, accompanied by a good practice of personal hygiene. Treatment of pregnant women in the second and third trimesters of pregnancy as well as children with acute brucellosis is carried out according to the “clinical protocol” (Table 8) [44]. In case of chronic brucellosis or cases in remote areas, the treatment course is supervised by the family medicine unit.

According to Dr. med. Aitkuluev N. (personal communication), many patients with acute brucellosis take broad-spectrum antibiotics on their own during fever which suppress the symptoms of acute brucellosis. After a few days they stop the treatment. However, after a few months (3–6 months), after hypothermia or hard physical work, these patients develop clinical manifestations without fever, e.g., polyarthritis of large joints, meningitis, clinical symptoms of hepatitis, endocarditis, endometritis, adnexitis, salpingoophoritis, orchiepididymitis, orchitis, and infertility. Miscarriage in pregnant women may also occur [45,46,47,48]. These patients see rheumatologists, neurologists, urologists, gynecologists, gastroenterologists, cardiologists, and receive ineffective treatment for several days. Failure of therapy at hospitals will result in a more in-depth examination including blood tests with Huddleson and Wright test titers of 1:50 or 1:100 in most cases.

## 10. Current Limitations and Potential for Improvement of Infection Control

In Kyrgyzstan, the trade of livestock and its products has increased in recent years, and is likely to continue to increase in the near future. The growth was rendered possible by stabilizing the epizootic situation together with an improvement of reproduction, i.e., better breeding strategies and herd maintenance. Also, compliance with veterinary and sanitary requirements and technical regulations of the EAEU as well as the introduction of electronic software (SITA, N(R)ADIS) to control the movement and diseases of animals helped with improving exports. In addition, since 2012, the decrease in brucellosis as a result of the above-mentioned vaccination campaigns, which were made possible with the help of the International Fund for Agricultural Development, had a positive impact on export numbers. Increase of production and decrease of the disease burden are considered the greatest achievements of the veterinary services in the last 10 years in Kyrgyzstan.

However, identification of strains and typing of Kyrgyz *Brucella* isolates have not been carried out by international reference laboratories yet, although molecular methods for subtyping isolates allow epidemiological surveillance, detection of the introduction of new strains and outbreaks investigation in brucellosis endemic regions [49]. For epidemiology purposes, strains from livestock and human populations must be genotyped and the corresponding techniques must be implemented immediately. Thus, “technical” epidemiology efforts must be strengthened now. International training on all aspects of brucellosis control need to be made available for Kyrgyz officers and scientists by OIE, WHO, FAO, and the EU or single states.

Currently, existing epidemiology methods and scientific research on brucellosis are very limited in Kyrgyzstan. Up to now no assessment of costs for the public health system or the state economy also considering days of work loss, reduced life span, and childcare costs is available. Lack of systematic research and representative data can make it challenging to implement more effective action plans on brucellosis. For this reason, the system needs serious modernization and strategic implementation of new approaches such as modern epidemiology and a digital health data information system. In addition, more routine data and epidemiological studies are needed, i.e., molecular epidemiology, case-control studies, and randomized multi-stage surveys, to gain a clear picture of the situation.

## 11. Discussion

The purpose of this study was to describe the epidemiological characteristics of human and animal brucellosis, its impact on animal production, and the role of implemented infection control measures during the period from 2010 to 2020 in Kyrgyzstan. Close cooperation of this disciplines was essential for this so that knowledge of historical most frequent problems and achievements can be shared with veterinarians, practitioners, and brucellosis researchers in neighboring countries, such as Uzbekistan, Tajikistan, Turkmenistan, and Kazakhstan. The regional and transregional context of livestock production in Kyrgyzstan highlights the need for close cooperation among the states involved to successfully combat brucellosis in Central Asia.

From 1994 to 2010, the number of human brucellosis cases and the incidence increased steadily throughout the country. During the period 2010–2020, data indicate that the presence of the disease has decreased steadily due to practices that reduced the risk of infection such as personal hygiene of private households and farms or more effective control measures. The highest incidences were mainly confirmed for Jalal-Abad, Talas, Naryn, and Issyk-Kul, where livestock farming is widely practiced and, therefore, the risk of infection is higher. Unfortunately, not all cases in humans are diagnosed and reported [50]. Thus, the true incidence of the disease is likely to be much higher than in official reports. The reasons for this situation are the reduced availability of medical facilities in some regions, the lack of specifically trained medical professionals and the lack of caution of people due to lack of knowledge about brucellosis and unspecific clinical symptoms of the disease [51]. There is poor public awareness of preventive measures in Kyrgyzstan. Therefore, most cases of brucellosis were registered in rural areas.

Acute brucellosis can progress to a chronic form with relapse, the development of persistent localized infections, or a nonspecific syndrome resembling the “chronic fatigue syndrome” [52]. During the study time, presumably, 1723 chronic cases resulted from self-medication during the acute stage. Moreover, a common cause of relapse (750 subacute cases) was an incomplete timespan of treatment if unwanted side effects occurred [53]. As recommended by the WHO, public health regulations in the country must be implemented to reduce overuse of antibiotics to prevent the development of (multi) resistant strains compromising the treatment regime. Such strains would also pose a trade impediment within EAEU since transmission to animal populations is possible.

Monitoring brucellosis cases in humans revealed a seasonality of the disease. Recent case-control studies have shown that direct contact with infected animals during lambing season or consumption of fresh cheese are the main sources of infection. Based on these findings, we assume that the high incidence of brucellosis in humans in late spring and early summer is indeed related to shearing and abortion during lambing time. Thus, brucellosis can be regarded as occupational disease and requires special measures to combat it taking this fact into account. Additional volunteer workers for the village health committees are needed to raise awareness for protective measures against brucellosis and to actively support vaccination of livestock in rural areas [54,55].

In case control studies, it was found that consumption of milk is not a risk factor for brucellosis, as milk is traditionally boiled before consumption. However, the consumption of cream/cheeses, which are prepared from untreated milk, is identified as a risk factor.

Summer pastures are often used in transhumance livestock farming on low and high lands [56]. This mobile livestock system avoids the overuse of lowland pastures and costly import of feed. In contrast, if more animals will be produced in the future, an intensive production system has to be introduced amending the traditional system, i.e., all year stable keeping and feed import [57]. Risk of spreading diseases will rise. Also, more and more contaminated manure is accumulating, posing a threat to the environment and wildlife if not disposed properly.

Kyrgyzstan has a responsibility to preserve endangered wildlife species. To protect them from disease, monitoring activities are necessary to register the epidemiological situation of brucellosis in wildlife. For this purpose, samples from wildlife, especially wild sheep can easily be collected during (commercial) hunting. An effective monitoring system will easily be available as part of a fair economical effort. Regardless of the low number of camels it is necessary to carry out control for brucellosis and other bacterial and viral diseases in these animals in the future to guarantee effective disease control in other farm animals [58].

In general, the system for brucellosis diagnosis, prevention, and control in the health and livestock sectors in Kyrgyzstan is well established. We believe that capacity building and greater international networking are needed to introduce new diagnostic and epidemiological methods and to provide training opportunities for young scientific staff. This training must include population-based data collection and interpretation, biosafety, and molecular microbiology. Further, harmonization between veterinary and human health sectors needs to be started regarding the applied test and protocols for brucellosis diagnosis.

## Figures and Tables

**Figure 1 microorganisms-10-01293-f001:**
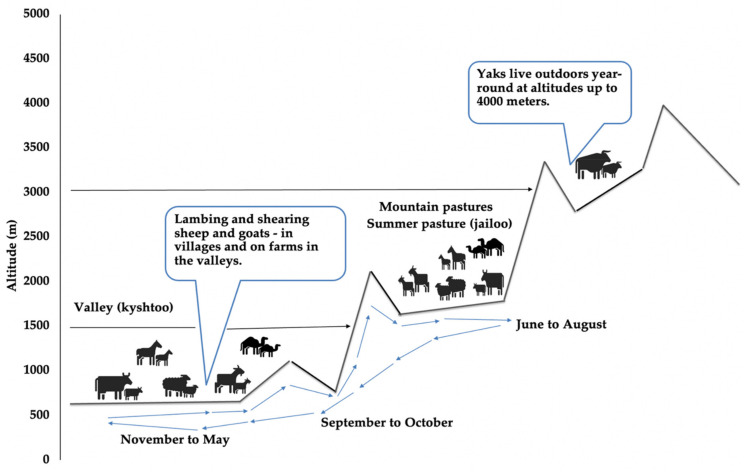
Transhumance seasonal movement of livestock in Kyrgyzstan. Adapted from Van Veen (1995) [12].

**Figure 2 microorganisms-10-01293-f002:**
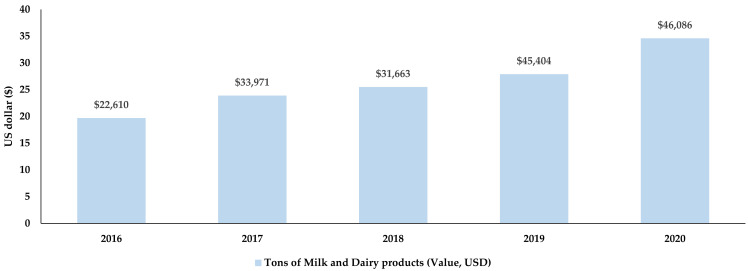
Export value of milk and dairy products from Kyrgyzstan from 2016 to 2020. Adapted from www.stat.kg, accessed on 11 January 2022 [20].

**Figure 3 microorganisms-10-01293-f003:**
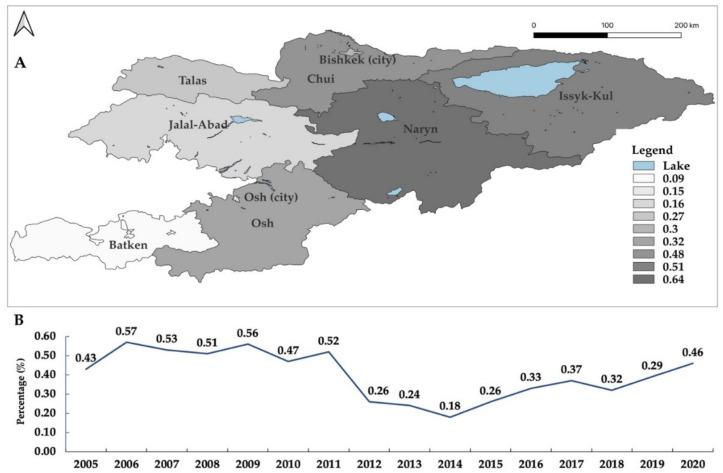
The seroprevalence of brucellosis in cattle by oblast in Kyrgyzstan from 2010 to 2020 (**A**). The seroprevalence of brucellosis in cattle from 2005 to 2020 (**B**). Cattle was positive in two consecutive Rose–Bengal plate test and/or serum agglutination test, based on reports from the SIVPS. The map was created using QGIS 3.22 Białowieża software which is available online [26].

**Figure 4 microorganisms-10-01293-f004:**
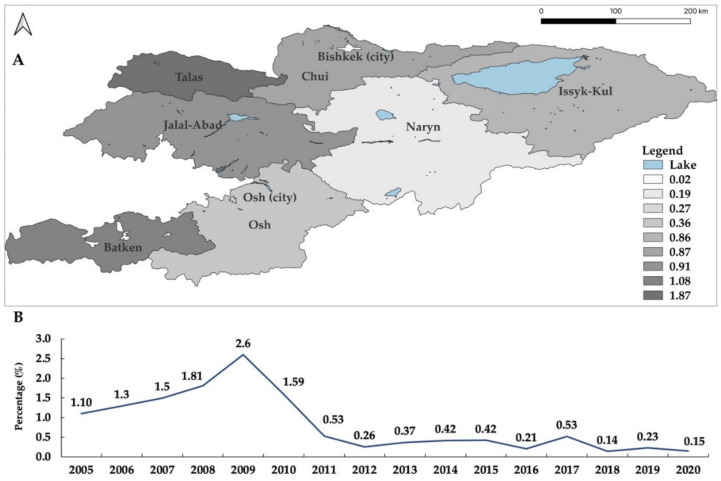
The seroprevalence of brucellosis in small ruminants by oblast in Kyrgyzstan from 2010 to 2020 (**A**). The seroprevalence of brucellosis in small ruminants 2005 to 2020 (**B**). Small ruminants were positive in two consecutive Rose–Bengal plate test and/or serum agglutination tests, based on reports of the SIVPS. The map was created using QGIS 3.22 Białowieża software which is available online [26].

**Figure 5 microorganisms-10-01293-f005:**
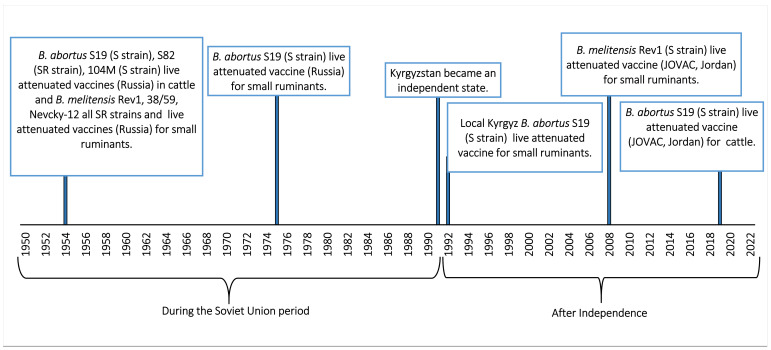
Timeline of vaccines used in Kyrgyzstan from 1950 to 2021. Adapted from Kasymbekov J. (2014), Ivanov A. V. (2011), Albertian M. P. (2006), personal communication Abdyraev M. (Duisheev KRIV); on 28 December 2021 [30,34,35].

**Figure 6 microorganisms-10-01293-f006:**
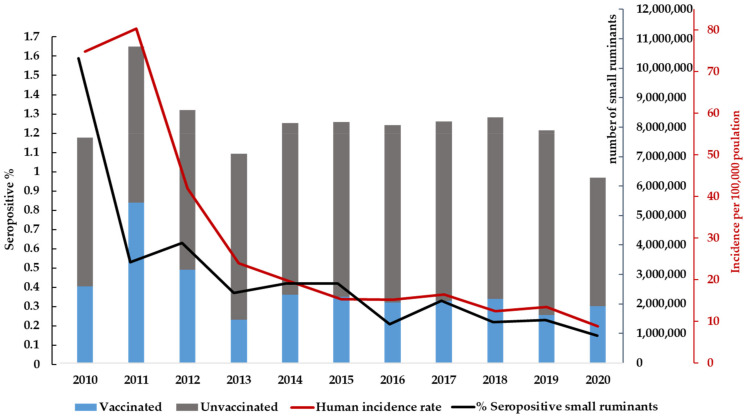
Number of vaccinated small ruminants in Kyrgyzstan from 2010 to 2020, based on data from RCQHDI and SIVPS.

**Figure 7 microorganisms-10-01293-f007:**
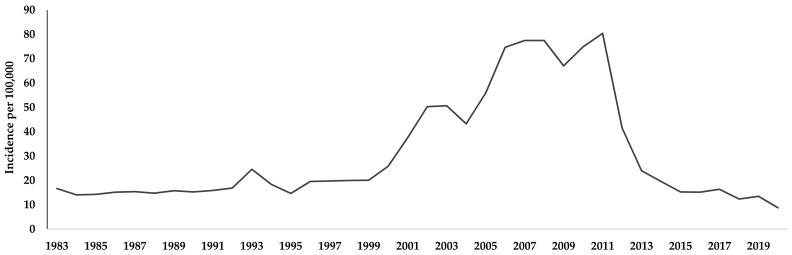
Incidence of human brucellosis in Kyrgyzstan from 1983 to 2019. Humans were tested positive in two consecutive Huddleson plate agglutination test and Wright tests, based on data from RCQHDI.

**Figure 8 microorganisms-10-01293-f008:**
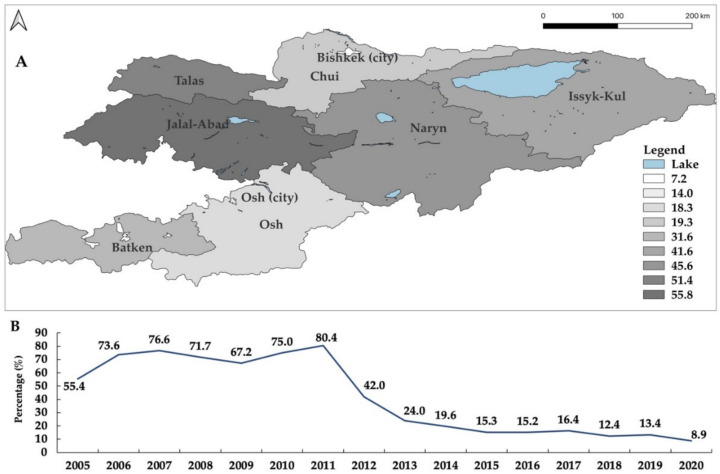
Incidence of human brucellosis by oblast in Kyrgyzstan from 2010 to 2020 (**A**). Incidence of human brucellosis from 2005 to 2020 (**B**). Humans were tested positive in two consecutive Huddleson plate agglutination and Wright tests. Adapted based on data from the RCQHDI. The map was created using QGIS 3.22 Białowieża software which is available online [26].

**Figure 9 microorganisms-10-01293-f009:**
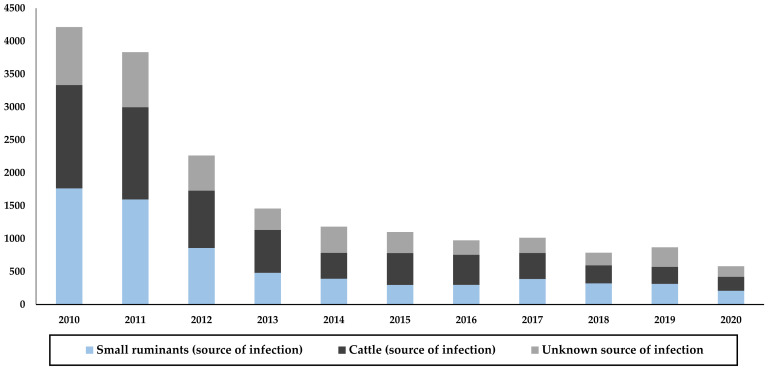
Sources of brucellosis infection in humans in Kyrgyzstan from 2010 to 2020, based on data from the RCQHDI.

**Figure 10 microorganisms-10-01293-f010:**
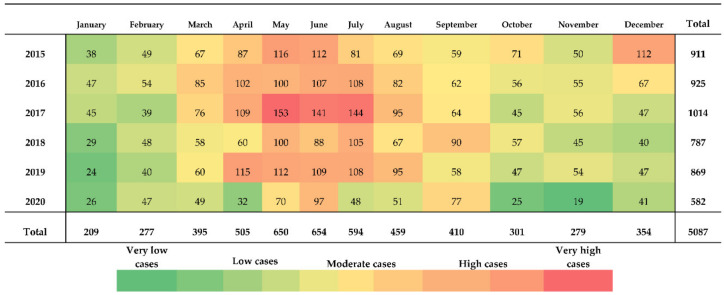
Reported cases of human brucellosis in Kyrgyzstan from 2015 to 2020 by per month. Numbers in bold give the sum of case numbers per year or month. Adapted based on data from the RCQHDI.

**Table 1 microorganisms-10-01293-t001:** Number of livestock in Kyrgyzstan from 2016 to 2020. Adapted from www.stat.kg, accessed on 3 February 2022 [13].

	2016	2017	2018	2019	2020
Sheep and goats	6,022,554	6,077,775	6,167,949	6,266,739	6,278,736
Cattle	1,527,763	1,575,434	1,627,296	1,680,750	1,715,776
Cows	769,933	789,796	812,596	835,270	855,050
Horses	467,249	481,329	498,684	522,611	539,644
Pigs	51,082	52,169	51,265	34,750	29,465
Poultry	5,673,607	5,910,418	6,009,697	6,211,184	6,070,443

**Table 2 microorganisms-10-01293-t002:** Export and import of animals and meat from/to Kyrgyzstan from 2016 to 2020 (Quantity * in thousands of heads animals and tons of meat). Adapted from www.comtrade.un.org, accessed on 3 January 2022 [15].

Countries	2016		2017		2018		2019		2020	
	Quantity *	Value, USD	Quantity	Value, USD	Quantity	Value, USD	Quantity	Value, USD	Quantity	Value, USD
Live Sheep and Goats (Foreign Economic Activity Commodity Nomenclature (FEACN) Code 0104) Export										
Kazakhstan	216	16,587	60	4590	25	1825	-	-	-	-
Tajikistan	1087	74,216	801	56,161	422	33,053	506	42,151	256	27,133
Afghanistan	-	-	440	35,339	-	-	-	-	-	-
Uzbekistan	-	-	-	-	440	110.000	2294	667,189	20,746	5,138,511
**Total**	$1303	$90,803	$1301	$96,090	$887	$144,878	$2800	$709,340	$21,002	$5,165,644
**Live sheep and goats (FEACN code 0104) import**										
Saudi Arabia	-	-	10	2867	-	-	-	-	-	-
Kazakhstan	-	-		-	13	1221				
Russian Federation	-	-	-	-	-	-	76	218	145	23,851
**Total**	0	0	10	$2867	13	$1221	76	$218	145	$23851
**Live bovine animals (FEACN code 0102) export**										
Kazakhstan	71	58,973	62	41,364	519	491,433	211	121,300	3704	6,202,245
Tajikistan	860	338,621	673	302,140	326	181,249	745	403,448	382	240,757
Uzbekistan	-	-	-	-	125	68,400	723	588,050	5705	8,010,703
Afghanistan	-	-	108	56,509	-	-	-	-	-	-
Azerbaijan	-	-	16	3319	-	-	-	-	-	-
Saudi Arabia	-	-	-	-	-	-	14	2800	-	-
**Total**	$931	$397,594	$859	$403,332	$970	$741,082	$1693	$1,115,598	$9791	$14,453,705
**Live bovine animals (FEACN code 0102) import**										
Uzbekistan	-	-	33	7684	-	-	-	-	-	-
Kazakhstan	-	-	-	-	77	9364	516	223,957	3387	1,843,410
Belarus	-	-	-	-	-	-	30	8601	556	247,478
Russian Federation	-	-	-	-	-	-	192	243,677	13,138	4,095,392
Austria	-	-	-	-	-	-	-	-	95	258,509
**Total**	$0	$0	$33	$7684	$77	$9364	$738	$476,235	$17,176	$6,444,798
**Sheep or goat meat, fresh, chilled or frozen (FEACN code 0204) export**										
Uzbekistan	-	-	-	-	-	-	-	-	9370	33,264
Iran	82,000	290,760	161,584	808,152	786,600	3,433,260	757,800	4,096,100	-	-
Qatar	1350	5000	-	-	-	-	-	-	-	-
United Arab Emirates	-	-	200	1400	150	50	3253	131,519	81,289	309,474
Kuwait	-	-	-	-	-	-	4617	1847	500	3000
Pakistan	-	-	-	-		-	0	358	-	-
**Total**	$83,350	$295,760	$162,984	$809,552	$786,750	$3,433,310	$807,223	$4,229,824	$91,159	$345,738
**Sheep or goat meat, fresh, chilled, or frozen (FEACN code 0204) import**										
Australia	37	403	-	-	-	-	-	-	-	-
New Zealand	76	1512	-	-	-	-	-	-	-	-
Mongolia	-	-	-	-	20,000	27,050	-	-	-	-
**Total**	$113	$1915	$0	$0	$20,000	$27,050	$0	$0	$0	$0
**Bovine meat, fresh or chilled (FEACN code 0201) and Frozen (FEACN code 0201) export**										
Kazakhstan	-	-	-	-	150	508	-	-	-	-
United Arab Emirates	-	-	-	-	-	-	630	2686	987	3645
India	-	-	-	-	-	-	-	-	84,000	168,000
Pakistan	-	-	-	-	-	-	-	-	18,000	129
**Total**	$0	$0	$0	$0	$150	$508	$630	$2686	$19,071	$171,774
**Bovine meat, fresh or chilled (FEACN code 0201) and Frozen (FEACN code 0201) import no data available**										
**Swine meat fresh, chilled or frozen (FEACN code 0203) export**										
Kazakhstan	-	-	1051	4337	-	-	-	-	-	-
**Swine meat fresh, chilled, or frozen (FEACN code 0203) import**										
China	373,000	393,500	-	-	-	-	-	-	-	-
Germany	56	244	-	-	2	30	-	-	-	-
Russian Federation	420,229	1,178,905	961,890	3,176,407	930,971	3,183,885	476,689	1,579,703	408,583	1,257,355
USA	54	385	-	-	-	-	-	-	-	-
Kazakhstan	-	-	75,804	276,419	20,720	60,011	-	-	-	-
Belarus	-	-	-	-	-	-	-	-	1293	3736
**Total**	$793,339	$1,573,034	$1,037,694	$3,452,826	$951,693	$3,243,926	$476,689	$1,579,703	$409,876	$1,261,091
**Live pigs (FEACN code 0203) export no data available**										
**Live pigs (FEACN code 0203) import**										
Czech Republic	-	-	105	95,634	-	-	-	-	-	-

**Table 3 microorganisms-10-01293-t003:** Import value of milk and dairy products (milk and cream: not concentrated, not containing added sugar or other sweetening matter (FEACN code 0401)) to Kyrgyzstan from 2016 to 2020 (Quantity * in tons of milk and dairy products). Adapted from www.comtrade.un.org, accessed on 3 January 2022 [15].

Countries	2016		2017		2018		2019		2020	
	Quantity *	Value, USD	Quantity	Value, USD	Quantity	Value, USD	Quantity	Value, USD	Quantity	Value, USD
Kazakhstan	499,557	450,159	631,152	524,437	711,455	456,314	743,637	410,473	3,510,929	2,144,728
Russian Federation	570,695	531,205	566,663	851,819	665,756	1,124,016	-	-	-	-
Tajikistan	1298	391	792	1265					-	-
Turkey	3933	8767	2040	1258	40	46	199	202	-	-
France			71	582					-	-
United Arab Emirates	-	-	46	50	-	-	-	-	-	-
Iran	-	-	-	-	-	-	18,700	14,220	-	-
Germany	1200	3233	-	-	-	-	-	-	-	-
Belarus	-	-	-	-	-	-	-	-	850	559
Total	$1,076,683	$993,755	$1,200,764	$1,379,411	$1,377,251	$1,580,376	$762,536	$424,895	$3,511,779	$2,145,295

**Table 4 microorganisms-10-01293-t004:** Current laws and Regulations of the Veterinary System in the Kyrgyz Republic. Adapted from Tilekeyev K. (2016), personal communication Sotovaldiev A. (principal adviser SIVPS); on 27 January 2022 [21].

Laws and Regulation	Brief Description
Law of the Kyrgyz Republic No. 175 “On veterinary practice” dated 30 December 2014	The law establishes the legal, social, financial, and economic basis for veterinary practice. It aims at protection of the population against diseases common to human beings and animals, ensuring animal welfare veterinary and sanitary safety for the national territory.
Priority veterinary and sanitary requirements for the prevention of animal diseases, approved by Decree of the Government of the Kyrgyz Republic No. 377 dated 18 June 2015	Approves the veterinary and sanitary requirements for the keeping, selling, and slaughtering of livestock, vaccination, diagnosis of infectious animal disease, livestock transport requirements, veterinary, and sanitary measures for the export of livestock, disinfection, pest control and deratization of facilities, and veterinary and sanitary protection requirements within the territories of the Kyrgyz Republic.
Law of the Kyrgyz republic No. 91 of 6 June 2013 “On the Identification of Animals” (As amended by the Law of the Kyrgyz Republic No. 131 of 12 July 2014)	The main objectives of this law are reliable counting of livestock, state veterinary control on exports and imports of animals and protection of the territory of the Kyrgyz Republic from the import of agents of contagious diseases, compliance of business entities with veterinary, sanitary, and zoohygienic requirements for keeping and breeding animals, trade in animals, slaughter, and disposal of unusable products of slaughter and dead animals.
Technical Regulations of the Customs Union (CU) No. 034/2013 “On the Safety of Meat and Meat Products” adopted by resolution of the Council of the Eurasian Economic Commission No. 68 dated 9 October 2013; effective from 1 May 2014	These technical regulations set the requirements for the safety of meat and meat products for production processes, storage, transportation, sale, and disposal, as well as requirements for labeling and packaging to protect human life and health and the environment of the territory of the Customs Union (CU).
Technical Regulations of the Customs Union (CU)No. 033/2013 “On the safety of milk and dairy products” adopted by resolution of the Council of the Eurasian Economic Commission No. 67 dated 9 October 2013, as amended on 10 July 2020	These technical regulations establish safety requirements for milk and dairy products of the territory of the Customs Union, for the processes of their production, storage, transport, sale and use, as well as requirements for labeling and packaging of milk and dairy products to ensure their free movement.
Law of the Kyrgyz Republic No. 88 “Technical Regulations:Food Production Hygiene” dated 1 June 2013	The law sets requirements for hygiene during production, processing, and storage of raw food materials and products, as well as for the technological processes and organization of production.
Resolution of the Government of the Kyrgyz Republic No. 385 dated 22 June 2015. About approval of the Concept of development of halal industries in the Kyrgyz Republic	Main objective is the determination of the main directions of development of halal industries in the Kyrgyz Republic, including promotion of domestic halal products for internal and foreign markets.

**Table 5 microorganisms-10-01293-t005:** Prevalence of brucellosis in Kyrgyzstan from 2010 to 2020 based on reports from the State Inspectorate for Veterinary and Phytosanitary Safety. Cattle and small ruminants were positive in two consecutive Rose–Bengal plate tests and/or serum agglutination tests, based on reports from SIVPS.

Years	Cattle				Small Ruminants			
	Number Cattle	Number Tested	Number Positive	% Positive from Tested	Number Small Ruminants	Number Tested	Number Positive	% Positive from Tested
2010	1,298,825	887,447	4171	0.47	5,037,715	204,403	3250	1.59
2011	1,385,830	1,056,731	5495	0.52	5,288,115	291,887	1547	0.53
2012	1,367,466	951,154	2473	0.26	5,423,881	34,127	215	0.63
2013	1,404,168	1,006,667	2416	0.24	5,641,214	29,189	108	0.37
2014	1,458,377	979,444	1763	0.18	5,829,024	30,714	129	0.42
2015	1,492,517	953,077	2478	0.26	5,929,529	124,762	524	0.42
2016	1,577,630	971,212	3205	0.33	6,022,554	32,381	68	0.21
2017	1,575,434	927,027	3430	0.37	6,077,775	30,755	163	0.53
2018	1,627,296	929,375	2974	0.32	6,167,949	28,636	41	0.22
2019	1,680,750	1,287,692	5022	0.39	6,266,739	29,130	67	0.23
2020	1,715,776	92,4816	4227	0.46	6,278,736	28,073	43	0.15

**Table 6 microorganisms-10-01293-t006:** Isolation sources of *Brucella* isolates in Kyrgyzstan. Adapted from [28,29,30,31,32].

Year	Location	*B. melitensis*		*B. abortus*		References
		Sheep and Goats	Cattle	Cattle	Sheep and Goats	
2011	Entire country	122	8	-	-	Chegirov S, 2014 [28].
2012	Ak-Tala rayon, Naryn oblast	285	2	2	19	Chegirov S, 2013 [29].
2013	Naryn oblast	15	2	-	-	Kasymbekov J et al., 2014 [30].
20016	Whole country	5	3	1	-	Atambekova Z et al., 2016 [31].
2018–2020	Naryn and Issyk-Kol oblasts	-	1 (yak)	8 (yaks)	-	Mambetali S et al., 2021 [32].
Total	Entire country	427	16	11	19	
		443		30		

**Table 7 microorganisms-10-01293-t007:** Prevalence of brucellosis in small ruminants with *B. melitensis* Rev1 vaccine in Kyrgyzstan from 1977–1989. Cattle and small ruminants were positive in two consecutive Rose-Bengal plate tests and/or serum agglutination tests. Based on reports from the SIVPS, adapted from Kasymbekov J. (2014) [30].

Year	Estimated Prevalence (%)		Abortions Due to Brucellosis	Number of Flocks with Abortions	Number of Human Cases
	Ewe	Ram			
1977	0.6	0.1	142	112	191
1978	1.0	0.1	169	120	215
1979	1.2	0.2	233	121	209
1980	1.3	0.2	269	127	221
1981	1.6	0.2	288	137	249
1982	1.9	0.4	407	141	292
1983	1.9	0.6	404	161	301
1984	1.9	1.0	409	197	327
1985	2.1	1.4	432	219	368
1986	2.3	1.5	555	241	411
1987	2.9	1.6	583	143	427
1988	3.3	1.6	599	260	489
1989	4.0	1.7	970	293	508

**Table 8 microorganisms-10-01293-t008:** Antibacterial therapy for pregnant women and children. Adapted from personal communication of Dr. Aitkuluev N. (principal adviser, the Republican Clinical Infectious Diseases Hospital; on 2 March 2022) and www.who.int/news-room/fact-sheets/detail/brucellosis, accessed on 12 February 2022 [44].

Patient	Standard Treatment Regimen	Alternative Medications
Children < 8 years	Doxycycline 100 mg twice daily followed by Rifampicin 600 mg per os once daily within 45 days.	Trimethoprim/Sulfamethoxazole 480 mg IV per os twice daily or Ciprofloxacin 0.5 g IM per os twice daily for 45 days and Gentamicin 80 mg twice daily for 45 days or Streptomycin 1.0 mg IM once or twice daily for 14 days
Children > 8 years	Trimethoprim/sulfamethoxazole 10 mg/kg/day IV per os twice daily and Rifampicin 8–10 mg/kg per os in 1or 2 doses within 45 days	Gentamicin 5 mg/kg/day for 45 days or Streptomycin 20–30 mg/kg IM once for 14 days
Pregnant women	Rifampicin 600 mg per os twice daily within 45 days	Trimethoprim/Sulfamethoxazole 480 mg IV per os twice daily for 45

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
