# Peer review of "Brucellosis in Humans and Animals in Kyrgyzstan"

_microorganisms, 2022, doi:10.3390/microorganisms10071293_

Round 1

Reviewer 1 Report

The authors described the situation of brucellosis in Kyrgyzstan in human and animals, so as the impact on animal production and the role of control measures in the country. This review is very interesting as it describes the situation in this country, so as the impact of mass vaccination in small ruminants. It is of importance to have this type of knowledge in many developing countries.

Just some comments / modifications:

Line 40: Brucella in italic

Line 58: the … can spread. Maybe a word (bacteria ?) is missing

Line 260: positive with which test?

Figure 5: 1954, 1992 and 2019, the end of the text is missing

Line 537: to describe

Line 540: in this review? (t seems to be missing)

Author Response

Thank for your interest and feedback on the manuscript. Accounting for the given suggestions, we have now revised and corrected the following points:  

  • Line 40: brucella in italic

Response: it has been corrected.

  • Line 58: the… can spread. Maybe a word (bacteria?) is missing

Response: we agree and have updated the sentence.

  • Line 260: positive with which test?

Response:  Thank you for the suggestion. We added the information (Rose Bengal plate test, serum agglutination test, Huddleson plate agglutination test and Wright test) (Table 5. Line 323; Table 7. Line 442; Figure 3. Line 351; Figure 4. Line 358; Figure 7. Line 445; Figure 8. Line 496.) to the text and the table caption.

  • Figure 5: 1954, 1992 and 2019, the end of the text is missing

Response: we have fixed the error.

  • Line 537: to describe

Response: we fixed the error.

Reviewer 2 Report

The manuscript entitled “Brucellosis in humans and animals in Kyrgyzstan” was reviewed. This is an interesting review dealing with a disease of public health and economic significance.

The manuscript needs to be edited for English language since grammatical and syntax errors are too many to be mentioned.

More importantly, there is a lot of general information (whole pages) given both for the disease itself and the agricultural sector in Kyrgyzstan which needs to be reduced.

To the contrary the information regarding the actual objective of the study should be more detailed. For instance, references in prevalence or incidence of the disease in humans and animals should be accompanied by the laboratory method applied.

Moreover, the significance of this review outside Kyrgyzstan should be presented, in other words, the authors should explain why this review is important for the international scientific readership.

On the contrary, the main strengths of this paper are figures which are very informative and easy to understand.

Author Response

We thank the reviewer for the constructive and thoughtful comments. We have addressed your comments as follows:

1). The manuscript needs to be edited for English language since grammatical and syntax errors are too many to be mentioned.

Response: We edited the language as demanded, hoping that now it reaches the standards.

2). More importantly, there is a lot of general information (whole pages) given both for the disease itself and the agricultural sector in Kyrgyzstan which needs to be reduced.

Response: We understand the reviewer's concern regarding the sections of the manuscripts which are not directly dealing with brucellosis as disease. However, we believe that the additional information, e.g. on import and export or on the legal background are important for understanding the risks of infection transmission and the possible legal measures that are involved. In that way, the reader can see the whole picture of brucellosis in Kirgizstan and which sectors have to be considered for improvement of brucellosis monitoring and control.  Thus, we tried to reduce the text while still including all information of the original manuscript.

3). To the contrary the information regarding are actual objective of the study should be more detailed. For intense, references prevalence or incidence of the disease in humans and animals should be accompanied by the laboratory method applied.

Response: We apologize for the lack of clarity regarding the objective. The objective of this study is now stated in more detail in the introductory part. (Line 85).

4). The incidence and prevalence of the disease in humans and animals have been indicated in the descriptions of the tables and figures according to the laboratory method used.

Response (Table 5. Line 323; Table 7. Line 442; Figure 3. Line 351; Figure 4. Line 358; Figure 7. Line 445; Figure 8. Line 496.)

5). Moreover, the significant of this review outside Kyrgyzstan should be presented, in other words, the authors should explain why this review is important for the international scientific readership.

Response: We apologize for the lack of information. The importance of this review is now stated in detail in the discussion part (Line 641).

Reviewer 3 Report

Dear authors,

This review describes different aspects of Brucellosis, important for both animals and humans, and in my opinion, it contains all the necessary information, it is easy-reading and well-structured. 

However, commas are lacking from a few sentences and abbreviations should be explained once in the text and then used. I believe that you can omit ''Kyrgyzstan'' when implied.

Author Response

Thank for your interest and feedback on the manuscript. We appreciate the Reviewer’s input and edited the manuscript according to the suggestions:

1). Commas are lacking from a few sentences and abbreviations should be explained once in the text and then used.

Response: We revised the entire text regarding the English language and commas. Abbreviations are explained once in the text and used in short form.

Round 2

Reviewer 2 Report

The authors have adequately responded to all my concerns. There is a considerable improvement in the quality of the paper including linguistic issues. The manuscript is better organized and the objective is now clearly stated.

Author Response

We are grateful to the reviewers for their insightful comments on our article!